# VTaC: A Benchmark Dataset of Ventricular Tachycardia Alarms from ICU Monitors

**Li-wei H. Lehman** *
MIT

**Benjamin Moody**
MIT

**Harsh Deep**
MIT

**Feng Wu**
MIT

**Hasan Saeed**
MIT

**Lucas McCullum**
MIT

**Diane Perry**
MIT

**Tristan Struja**
MIT

**Qiao Li**
Emory University

**Gari Clifford**
Emory University, Georgia Tech

**Roger G. Mark**
MIT

## Abstract

False arrhythmia alarms in intensive care units (ICUs) are a continuing problem despite considerable effort from industrial and academic algorithm developers. Of all life-threatening arrhythmias, ventricular tachycardia (VT) stands out as the most challenging arrhythmia to detect reliably. We introduce a new annotated VT alarm database, VTaC (**V**entricular **T**achycardia annotated **a**larms from I**C**Us) consisting of over 5,000 waveform recordings with VT alarms triggered by bedside monitors in the ICUs. Each VT alarm in the dataset has been labeled by at least two independent human expert annotators. The dataset encompasses data collected from ICUs in three major US hospitals and includes data from three leading bedside monitor manufacturers, providing a diverse and representative collection of alarm waveform data. Each waveform recording comprises at least two electrocardiogram (ECG) leads and one or more pulsatile waveforms, such as photoplethysmogram (PPG or PLETH) and arterial blood pressure (ABP) waveforms. We demonstrate the utility of this new benchmark dataset for the task of false arrhythmia alarm reduction, and present performance of multiple machine learning approaches, including conventional supervised machine learning, deep learning, contrastive learning and generative approaches for the task of VT false alarm reduction.

## 1 Introduction

Within intensive care units (ICUs), bedside monitors generate a substantial number of alarms, a considerable proportion of which are false [Drew et al., 2014]. These false alarms not only burden clinicians with heightened cognitive strain but also have the potential to obscure genuine alarms, thereby endangering patient safety. Arrhythmia alarms, accounting for 45% of overall ICU alarms, in particular, pose a significant challenge, contributing to alarm fatigue that increases the risk of healthcare providers potentially overlooking true life-threatening events [Drew et al., 2014, Cvach, 2012].

Accurate and reliable algorithms that can distinguish between true vs. false arrhythmia alarms can improve the overall effectiveness of monitoring systems in ICUs. However, the development and evaluation of such algorithms rely on the availability of high-quality and representative datasets with ground-truth labels from human experts.

---

*Corresponding author: lilehman@mit.edu.

37th Conference on Neural Information Processing Systems (NeurIPS 2023) Track on Datasets and Benchmarks.

We focus on the problem of detecting false alarms for one of the life-threatening arrhythmias, ventricular tachycardia (VT), defined as five or more ventricular beats with a heart rate higher than 100 bpm (beats-per-minute) [Clifford et al., 2015a, 2016]. Among all the life-threatening arrhythmia alarms, VT alarms are one of the most commonly occurring alarms [Drew et al., 2014, Aboukhalil et al., 2008] and false VT alarms have proven to be the most challenging to detect reliably [Aboukhalil et al., 2008, Clifford et al., 2015a, 2016, Zhou et al., 2022].

In this study, we present a new annotated VT database designed to address the challenges associated with false arrhythmia alarm reduction. The dataset consists of a total of over 12,000 labeling decisions from six experts who reviewed over 5,000 waveform recordings with VT alarms triggered by patient bed-side monitors in an ICU setting. These records were sourced from multiple ICUs from three major hospitals in the United States, providing a diverse and representative collection of waveform data. Each waveform recording comprises electrocardiogram (ECG) leads and one or more pulsatile waveforms, such as photoplethysmogram (PPG or PLETH) and arterial blood pressure (ABP) waveforms.

One important aspect of our dataset is the inclusion of labeling decisions from multiple independent human annotators. This approach helps to ensure the robustness of the dataset by reducing the potential impact of individual subjectivity and variability among annotators. The availability of such a comprehensive and annotated dataset enables researchers and practitioners to evaluate and compare the performance of various machine learning approaches for false arrhythmia alarm reduction. We demonstrate the utility of our benchmark dataset by evaluating multiple machine learning approaches, including conventional supervised machine learning, deep learning, contrastive learning, and generative approaches.

By addressing the challenges of false arrhythmia alarms and providing a benchmark dataset, this research contributes to the ongoing efforts to improve patient safety, reduce alarm fatigue, and enhance the efficiency of monitoring systems in ICUs. The introduced dataset VTaC, to the best of our knowledge, represents the largest open-access, multi-center VT alarm dataset, meticulously annotated by highly skilled clinical experts. It provides a valuable resource for developing algorithms to reduce false VT alarms, a longstanding and unsolved challenge in ICUs. Data will be available on PhysioNet at www.physionet.org/content/vtac/.

## 2    Background and Machine Learning Challenges

In this section, we outline challenges in developing machine learning approaches for false VT alarm reduction.

**Artifact or noise.** Erroneous triggers of VT alarms typically occur due to noise and ECG-related artifacts, such as electrode movements, poor electrode contact, or other technical reasons. Ventricular arrhythmias are characterized by distortion of beat morphology with a broader QRS complex. However, a challenge to be overcome is that noise and artifacts can exhibit morphologies similar to abnormal QRS complexes, almost indistinguishable from the periodic anomalies from a true VT episode [Clifford et al., 2015a].

**Other rhythms with similar appearance.** VT and other rhythms, such as atrial fibrillation (A-Fib) and aberrantly conducted supra ventricular tachycardia (SVT) may exhibit similar characteristics on an ECG, making it difficult to distinguish them [Littmann et al., 2019]. For example, both VT and SVT with aberrant conduction can have wide QRS complexes on the ECG. In some cases, patients may have multiple simultaneous rhythms, further complicating the interpretation. For instance, VT may coexist with A-Fib or SVT, making it challenging to identify and differentiate each rhythm component.

**Individual differences and variable morphology.** VT can have variable morphologies, which means it can appear differently in different patients or even within the same patient. This variability can make it challenging to identify VT solely based on ECG characteristics immediately prior to the alarm onset. Inspection of the same individual patient's prior ECG characteristics at baseline may be necessary.

**Long sequence with complex patterns and Missing Channels.** In the false alarm reduction problem, one major challenge stems from the long time-series sequence and the sparse availability of labeled data [Clifford et al., 2015a, Kiyasseh et al., 2021a, Zhou et al., 2022]. At a sampling frequency of

250 Hz, a 5-minute segment of a single channel of ECG contains 75000 samples. The label of each record, on the other hand, is just TRUE and FALSE, providing limited supervised information to train a deep learning model. Additionally, missing channels pose a significant challenge for false alarm reduction.

**Sparse availability of high-quality labeled data.** While machine learning and deep learning have made significant advances in many domains, including image and voice analyses, the application of deep learning in physiological waveform analysis has had limited success, partly due to limited availability of high-quality labeled data. The best performing approaches for false arrhythmia alarm detection in the 2015 PhysioNet Challenge rely on a combination of expert-defined rule-based logic analysis [Plesinger et al., 2015] and simple machine-learning models, while, in comparison, conventional deep learning approaches generally under-performed in the 2015 PhysioNet Challenge [Clifford et al., 2015a].

## 3 Related Works

### 3.1 Related Works in Annotated Arrhythmia Datasets

Several efforts have been made in the development and curation of annotated arrhythmia datasets, which serve as valuable resources for the development and evaluation of algorithms aimed at arrhythmia detection and classification. These datasets contribute to advancing the field of cardiac monitoring and have facilitated the progress of machine learning techniques in arrhythmia analysis. In this section, we highlight some notable related works in the domain of annotated arrhythmia datasets.

One seminal dataset widely used in arrhythmia research is the MIT-BIH Arrhythmia Database [Moody and Mark, 2001]. It comprises 48 ECG records (each slightly more than 30 minutes) obtained from long-term Holter recordings. These records include a mix of randomly selected and specifically chosen cases, showcasing a broad spectrum of typical clinical waveforms, artifacts, and complex arrhythmias, both ventricular and supraventricular. The dataset has been instrumental in benchmarking different algorithms for arrhythmia classification and contributed to the evaluation of signal processing and machine learning techniques. The MIT-BIH Arrhythmia Database does not include annotations for arrhythmia alarms generated by commercial bed-side monitors.

The 2015 PhysioNet Challenge event [Clifford et al., 2015b, 2016] focused on five types of life-threatening arrhythmias, including ventricular tachycardia, asystole, extreme bradycardia, extreme tachycardia, and ventricular fibrillation/flutter. The goal of the Challenge was to reduce the number of false alarms, while avoiding suppression of true alarms. The Challenge consists of two events: (1) real-time classification using only data up to the alarm onset; (2) retrospective analysis in which the contestants are allowed to use the 10-second data after the alarm onset for classification.

The 2017 PhysioNet/CinC Challenge [Clifford et al., 2017] is a collection of single-lead ECG recordings (each between 30 to 60 seconds) from wearable devices. The primary objective of this dataset is to facilitate the development of algorithms for the classification of cardiac rhythms into four main categories: normal sinus rhythm, atrial fibrillation (AF), an alternative rhythm, or noisy recordings that cannot be classified.

Most recently, [Pelter et al., 2023] introduced the UCSF VT dataset comprising 18,455 annotated VT alarms from 858 ICU patients at UCSF. Their research differs from ours in several key aspects. The UCSF VT dataset was exclusively collected from a single hospital, utilizing patient monitors from a single commercial bed-side monitoring vendor. In contrast, VTaC comprises a multi-center collection, incorporating data from monitors manufactured by three major vendors of commercial monitoring systems. As a result, VTaC provides a more diverse and comprehensive representation, which is crucial for the development of VT false alarm reduction algorithms. Furthermore, the UCSF VT dataset, as indicated by [Pelter et al., 2023], is not shared or made publicly available to the research community. Conversely, VTaC is an open-access resource, freely accessible to the research community.

### 3.2 Machine Learning Approaches to False Arrhythmia Alarm Reduction

Multiple algorithms and machine learning approaches have been proposed for arrhythmia analyses and false alarm reduction. Conventional algorithms [Aboukhalil et al., 2008, Li and Clifford, 2012]

and the best performing entries in the 2015 PhysioNet Challenge for false arrhythmia alarm reduction [F Plesinger et al., 2016, Kalidas and Tamil, 2015, Clifford et al., 2015a] rely on signal processing and expert-defined rule-based logic to analyze physiological signals, or using manually engineered features as inputs for machine learning classifiers. Due to the sparse availability of labeled data, much of the recent machine learning approaches for false alarm reduction focus on techniques that enable label-efficient learning, using e.g. multi-task learning [Schwab et al., 2018], supervised representation learning [Lehman et al., 2018], and self-supervised or contrastive learning [Kiyasseh et al., 2021b, Zhou et al., 2022]. Lehman et al. [2018] focus on VT false alarm detection, and use FFT-transformation of individual ECG beats for scalable learning. Kiyasseh et al. propose a family of self-supervised pretraining mechanisms based on contrastive learning for physiological signals[Kiyasseh et al., 2021b]. Zhou et al. [2022] proposed a contrastive learning framework to reduce five types of life-threatening arrhythmia alarms. More recently, Wu et al. [2023] proposed a conditional generative modeling approach for the task of VT false alarm reduction, and showed promising results from a diffusion contrastive learning model using an annotated arrhythmia dataset [Aboukhalil et al., 2008] from the MIMIC II database [Saeed et al., 2011].

## 4    Data Collection and Annotation Methodology

In this section, we briefly outline our data collection methodology and the annotation process. For detailed description, please refer to the Appendix.

### 4.1    Data

We extracted and compiled a total of 18,465 waveform VT alarm events, corresponding to 2,376 unique patient waveform records from bed-side monitors from three leading commercial vendors. These records were sourced from multiple ICUs from three major hospitals in the United States, providing a diverse and representative collection of waveform data. Each waveform recording in our dataset consists of a 10-minute segment that encompasses the onset of the ventricular tachycardia (VT) alarm. This segment includes 5 minutes of waveform data preceding the alarm onset and 5 minutes following it. To maintain diversity, we randomly selected a maximum of five alarm events from any particular patient waveform record, yielding a total of 5,742 events for annotation. This ensures that a reasonable number of events were sampled from different patient records, preventing an over-representation of events from any single patient record. Waveform records were de-identified to ensure the anonymization of all identifiable information, including patient names, dates, and medical record numbers. All signals were resampled to 250 Hz, and all signal labels were adjusted to match the nomenclature used in the PhysioNet Challenge 2015 database.

### 4.2    Annotation Process

Following the PhysioNet Challenge 2015, a VT episode is defined as five or more consecutive ventricular beats with heart rate higher than 100 beats-per-minute (bpm) [Clifford et al., 2015a]. Each event was reviewed and labeled by at least two annotators independently. For our task, annotators were given the options of "True" for when they believe the alarm was correct, "False" for when they believe the alarm was incorrect, "Uncertain" for when they were unsure which annotation to assign, "Reject" for when the alarm was unreadable due to noise, artifacts, or other reasons. In order to reconcile conflicts between two annotator decisions, an adjudication process was implemented to resolve the conflicts. These disagreements were resolved either through direct one-on-one discussions between the annotators involved or by an adjudicator's vote to break the tie.

A total of 5,742 events were annotated by at least two independent annotators. Two independent annotators reached unanimous decisions on 4,534 (78.96%) events, whereas 21.04% (N=1,208) of the events received conflicting labeling decisions by two human annotators. Among the events with conflicting decisions, 816 (66.55%) were adjudicated. After removing 392 un-adjudicated events, a total of 5,350 alarm events received final labeling decisions. See Table 5 in Appendix for the breakdown of the 5,742 events based on the annotation decisions.

**Data Composition by Labeling Decisions** Table 1 lists a summary of alarms categorized by annotation decisions among the 5,350 alarms with final labeling decisions. A total of 102 (1.91%) and 211 (3.94%) events received Uncertain and Reject decisions respectively. After excluding "Rejected"

Table 1: Composition of the alarms categorized by labeling decisions.

|  | Unanimous | Adjudicated | Total |
|---|---|---|---|
| Total Alarm Events | 4,534 | 816 | 5,350 |
| Uncertain | 19 | 83 | 102 |
| Reject | 120 | 91 | 211 |
| True (T) | 1,222 | 219 | 1,441 |
| False (F) | 3,173 | 423 | 3,596 |
| **Final (T/F) Events Included** | **4,395** | **642** | **5,037** |
| Percent True Alarms | 27.80% | 34.11% | 28.61% |

and "Uncertain" events, the final dataset used for modeling contains 5,037 events, among which 4,395 events had unanimous annotation decisions, and the final labeling decisions of the 642 adjudicated events were based on the final decisions from the adjudicators. Figure 2 in the Appendix A illustrates the VTaC data collection and event sampling process.

**Annotation Team** The annotation team consists of six annotators, including a highly-experienced board certified cardiac arrhythmia technician, and a leading arrhythmia analysis expert physician who built the MIT-BIH Arrhythmia database. The team also includes three clinicians, and one biomedical signal processing engineer specializing in arrhythmia. We describe below expertise of three of the annotators who had contributed the most number of annotations. **Annotator 1**: Board certified cardiac arrhythmia technician with over 40 years of experience in interpreting and annotatingarrhythmias from ECG recordings. **Annotator 2**: Developed the MIT-BIH Arrhythmia database; physician in internal medicine with decades of experience in real-time arrhythmia analyses and clinical interpretation of Holter recordings. **Annotator 3**: Endocrinologist and Internist with extensive experience in Emergency Medicine and cardiovascular intensive care. Additional details of the annotation team can be found in the Appendix.

**Example True vs False VT Alarms in VTaC** In Figure 1, we show waveform plots for two example VT alarms that have been labeled as a true vs. a false alarm respectively. In each figure, waveform recordings from the final 10-seconds prior to the VT alarm onset were shown.

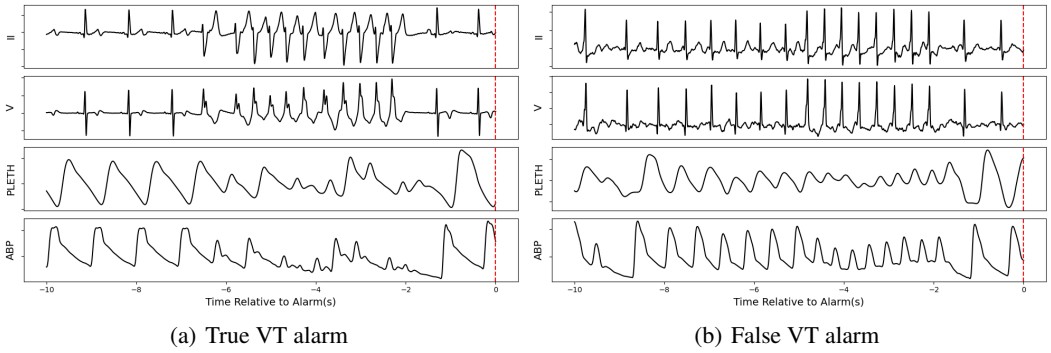

(a) True VT alarm          (b) False VT alarm

Figure 1: Example true vs. false VT alarms. Each plot shows data in the 10-second interval immediately prior to the VT alarm onset. The alarm onset is marked with a vertical red-line at time 0. Figure (b) shows an example false VT alarm – the event corresponds to an episode of atrial fibrillation with rate-related aberration instead of a ventricular tachycardia.

## 4.3 Comparison with Other Annotated VT Databases

In Table 2, we compare our newly developed VT alarm dataset with existing annotated VT databases on PhysioNet, including the training and hidden test sets from the PhysioNet Challenge 2015 [Clifford et al., 2015a], and an annotated arrhythmia alarm dataset from MIMIC II [Saeed et al., 2011, Goldberger et al., 2000] as reported in Aboukhalil et al. [2008]. We compare these datasets in

terms of annotated alarm events, the percentage of true alarms, as well as percentage of alarm events with ECG, ABP, and PLETH. We note that VTaC is a federated database sourced from waveform recordings collected from multiple commercial bed-side monitors (as such, multi-vendor) across multiple ICUs in three US hospitals. We also note that the current dataset provides fine-grained pre-adjudication labeling decisions from human experts, which are not included in prior datasets. As we randomly select VT alarm events from available recordings in constructing our dataset, the federated VT alarm dataset contains a false alarm rate that closely reflects the actual false alarm rate in a real-world ICU setting [Drew et al., 2004].

Table 2: Comparison of Annotated VT Databases on PhysioNet. Pre-Adj Decisions indicate whether labeling decisions before the adjudication processes are available.

|  | Challenge'15 Train | Challenge'15 Test | MIMIC II | VTaC |
|---|---|---|---|---|
| N (# VT alarms) | 341 | 221 | 1,900 | 5,037 |
| % True | 26.67% | 20.36% | 53.40% | 28.61% |
| Open Access* | Y | N | Y | Y |
| Multi-Vendor | Y | Y | N | Y |
| Pre-Adj. Decisions | N | N | N | Y |
| ECG (% events) | 100% | 100% | 100% | 100% |
| ABP (% events) | 54% | 57% | 100% | 36% |
| PLETH (% events) | 83% | 81% | 0% | 91% |

## 5   Models & Evaluation

We demonstrate the utility of this new benchmark dataset for the task of false VT alarm reduction, and present performance of multiple machine learning approaches in both real-time and retrospective settings following PhysioNet Challenge 2015 [Clifford et al., 2015a]. In the real-time setting, the goal is to reduce false alarms in "real-time" without using information after the alarm onset. In the retrospective setting, algorithms are allowed to utilize data within a brief time interval following the alarm, such as the 30-second interval specified in the PhysioNet Challenge 2015. However, in the context of this study, the retrospective setting incorporates data up to 5 seconds after the alarm.

### 5.1   Dataset

The final dataset used for modeling consists of 5,037 annotated VT alarms. The channels used for modeling included ECG, ABP, and PLETH (or PPG). We employed an 80-10-10 split for the train, validation, and test sets. The splits were performed at the patient record level rather than the individual alarm events level to ensure that events belonging to the same patient record are not distributed across the train, validation, and test sets. Table 3 displays the distribution of true alarms and signal types across the train, validation, and test set.

For the real-time event, 10-second window of waveform data immediately prior to the alarm onset were used as input to all of our models, with the exception that the generative models used a smaller window of data due to computational constraints. The retrospective events used 15-second window of waveform data to include the additional 5-second window of data immediately after the alarm onset. For each VT event, the model was fed with a total of four channels of waveform data, comprising two ECG leads, one ABP, and one PPG, all at a frequency of 250 Hz.

**Data Preprocessing** For ECG, we perform the following filtering: 1) a high-pass filter with 1-Hz cutoff frequency to suppress residual baseline wander; 2) a second-order 30 Hz Butterworth low-pass filter to reduce high frequency noise; and 3) a notch filter to eliminate power line interference. For PPG signal, we utilize a high-pass filter with a stopband frequency of 0.3 Hz and a passband frequency of 0.5 Hz, along with a low-pass filter with a passband frequency of 5 Hz and a stopband frequency of 8 Hz. We conduct z-normalization for all signals by utilizing the mean and standard deviation of each individual signal segment before feeding them into the model. In cases where signals are missing, we impute them with a value of 0.

Table 3: Composition of the train, validation and test splits. Unanimous decisions had two annotators who were both in agreement.

|  | Overall | Train | Validation | Test |
|---|---|---|---|---|
| Alarms (count) | 5,037 | 4,060 | 495 | 482 |
| Records (count) | 2,260 | 1,808 | 226 | 226 |
| True Alarms (count) | 1,441 | 1,163 | 141 | 137 |
| % True | 28.61% | 28.65% | 28.90% | 28.48% |
| ECG (% events) | 100% | 100% | 100% | 100% |
| ABP (% events) | 35.7% | 35.6% | 36.5% | 35.6% |
| PLETH (% events) | 90.6% | 90.42% | 89.83% | 92.32% |
| Adjudicated | 642 | 542 | 46 | 54 |
| Unanimous | 4,395 | 3,518 | 449 | 438 |

## 5.2 Models

**Models and Baselines.** We compare the performance of the following baseline and models. 1) **Rule-Based Method:** For the rule-based approach, we used the implementation from Plesinger et al. [2015], a winning entry in the PhysioNet 2015 challenge for false arrhythmia alarm reduction. Plesinger et al. [2015] test each channel in the record for regular heart activity using the QRS detection and derived R-R information [Plesinger et al., 2015]. 2) **MLP**: We apply the multi-layer perceptron as a feature extractor of the input waveform and then use a dense layer to classify. 3) **SAE** (supervised autoencoder): We use autoencoder to learn a lower dimensional representation from 10-second waveform signals, and simultaneously minimize the reconstruction loss from autoencoder and the cross entropy loss as described in [Lehman et al., 2018]. One notable difference from [Lehman et al., 2018] is that we utilize the entire 10-second waveform segment as the model input, without heartbeat detection for learning beat-by-beat representations. 4) **Transformers:** We utilize the Transformer encoder [Vaswani et al., 2017] as the feature extractor of the input waveform. 5) **CNN**: We use a light-weight 1-D convolutional neural network as a feature extractor [Zhou et al., 2022]. 6) **CNN+CL**: We use a light-weight 1-D convolutional neural network as a feature extractor and augment it with contrastive learning [Zhou et al., 2022]. 7) **FCN**: We use a fully-connected convolutional network as the feature extractor of the input waveform [Zhou et al., 2022]. 8) **FCN+CL**: We use a fully-connected convolutional network as the feature extractor of the input waveform and augment it with contrastive learning [Zhou et al., 2022]. 9) **BeatGAN**: BeatGAN [Zhou et al., 2019] is an GAN-based unsupervised anomaly detection algorithm for time series data. 10) **TanoGAN**: TanoGAN [Bashar and Nayak, 2020] is a novel GAN-based unsupervised method for detecting anomalies when a small number of data points are available. 11) **Diffusion+CL**: we use a diffusion model to reconstruct trajectories of physiological signal and classify alarms based on distances between the reconstructed and the actual trajectories [Wu et al., 2023].

## 5.3 Experiments

### Training

For the supervised approaches, including their variants with contrastive learning, we conducted model training for a maximum of 500 epochs. For the real-time events, hyperparameters were chosen through a grid search, and the Adam optimizer [Kingma and Ba, 2014] with a weight decay of 0.005 was employed to minimize both binary cross-entropy (BCE) loss and the discriminative constraint. Hyperparameter tuning was carried out, and a comprehensive description of the hyperparameter settings for all models can be found in the Appendix. For each machine learning model, we identified the hyperparameter setting with the best validation performance and proceeded to train the model with that setting using 10 different random seeds. We report the mean and standard deviation from the 5 random seeds with best validation performance. Model selection was based on the best Challenge Score on the validation set.

For retrospective events, we used the optimal hyper-parameter setting from the grid search of the real-time event, and proceeded to train the model with that setting using 10 different random seeds.

Table 4: Real-Time event performance from the VTaC database. Mean and standard deviation from 5 seeds shown. Top-performing models for each specific metric highlighted in bold. CL=Contrastive Learning.

| Methods | | TPR | TNR | PPV | F1 | Score | AUC |
|---|---|---|---|---|---|---|---|
| Rule-based | | **0.942** | 0.629 | 0.502 | 0.655 | 67.32 | – |
| Supervised | MLP | $0.597_{\pm0.087}$ | $0.691_{\pm0.09}$ | $0.441_{\pm0.037}$ | $0.502_{\pm0.015}$ | $45.58_{\pm1.10}$ | $0.706_{\pm0.008}$ |
| | SAE | $0.848_{\pm0.028}$ | $0.790_{\pm0.032}$ | $0.617_{\pm0.028}$ | $0.713_{\pm0.012}$ | $68.77_{\pm1.11}$ | $0.896_{\pm0.007}$ |
| | Transformer | $0.837_{\pm0.039}$ | $0.707_{\pm0.060}$ | $0.535_{\pm0.048}$ | $0.651_{\pm0.030}$ | $62.73_{\pm2.78}$ | $0.852_{\pm0.006}$ |
| | CNN | $0.928_{\pm0.006}$ | $0.782_{\pm0.019}$ | $0.629_{\pm0.019}$ | $0.750_{\pm0.013}$ | $76.17_{\pm1.20}$ | $0.936_{\pm0.009}$ |
| | FCN | $0.920_{\pm0.025}$ | $\mathbf{0.855}_{\pm0.018}$ | $\mathbf{0.717}_{\pm0.024}$ | $\mathbf{0.805}_{\pm0.016}$ | $\mathbf{80.08}_{\pm2.46}$ | $\mathbf{0.949}_{\pm0.006}$ |
| Supervised+CL | CNN+CL | $0.930_{\pm0.016}$ | $0.823_{\pm0.012}$ | $0.676_{\pm0.012}$ | $0.783_{\pm0.003}$ | $79.07_{\pm0.99}$ | $0.943_{\pm0.005}$ |
| | FCN+CL | $0.931_{\pm0.028}$ | $0.811_{\pm0.046}$ | $0.666_{\pm0.048}$ | $0.775_{\pm0.024}$ | $78.41_{\pm0.87}$ | $0.932_{\pm0.008}$ |
| Generative | BeatGAN | $0.597_{\pm0.100}$ | $0.597_{\pm0.091}$ | $0.373_{\pm0.033}$ | $0.455_{\pm0.037}$ | $41.02_{\pm2.71}$ | $0.597_{\pm0.028}$ |
| | TAnoGAN | $0.704_{\pm0.053}$ | $0.611_{\pm0.076}$ | $0.421_{\pm0.027}$ | $0.524_{\pm0.008}$ | $47.61_{\pm0.87}$ | $0.657_{\pm0.012}$ |
| | Diffusion+CL | $0.853_{\pm0.054}$ | $0.517_{\pm0.040}$ | $0.412_{\pm0.013}$ | $0.555_{\pm0.016}$ | $52.51_{\pm2.61}$ | $0.685_{\pm0.017}$ |

**Evaluation** The evaluation metrics for false alarm reduction are true positive rate (TPR), true negative rate (TNR), positive predictive value (PPV), F1-score, and area under the ROC (receiver operating characteristic) curve (AUC). The PhysioNet Challenge 2015 also provides an official scoring mechanism for evaluating [Clifford et al., 2015a]. It is defined as $score = (TP + TN)/(TP + TN + FP + 5 * FN)$, where $TP$ is true positives, $FP$ is false positives, $FN$ is false negatives, and $TN$ is true negatives. The Challenge Score focuses more on the TPR value, since mistakenly classifying a true alarm as false results in significantly more severe consequences. The PPV, F1, TPR, TNR and Challenge Score are determined based on a threshold value of predicted probability that maximizes the validation Challenge Score as a cut-off value to determine true vs. false alarms in the test set.

## 6 Performance in Real-Time VT Alarm Classification

In this section, we present the experimental results of eleven models for the real-time event as outlined in Table 4. Results for the retrospective event are presented in the Appendix Table 8. The winning entry of the 2015 PhysioNet challenge using a Rule-based approach serves as a reference point for benchmarking, achieving a Challenge Score of 67.32 in the real-time event.

Our findings indicate that both FCN and CNN, along with their contrastive learning variants, demonstrated superior performance when compared to other models. Notably, FCN emerged as the top performer, achieving the highest Challenge Score (80.08±2.46) and AUC (0.949±0.006), surpassing other models. Contrastive learning enhanced the performance of 1-D CNN, as evidenced by a notable improvement in the Challenge Score from 76.17 to 79.07. However, this performance boost was not observed in the case of FCN.

Our study reveals distinctive findings when contrasted with Zhou et al. [2022]. In their investigation, the application of FCN to a considerably smaller dataset from the 2015 PhysioNet Challenge resulted in significantly inferior performance compared to the lightweight 1-D CNN and CNN+CL. In contrast, our analysis, conducted on the VTaC dataset, demonstrated that FCN outperformed 1-D CNN in terms of AUCs and Challenge Scores. This difference in outcomes could potentially be attributed to the common observation that increasing the size of the labeled dataset often leads to improved performance in more complex models.

Generative approaches that previously demonstrated superior performance [Wu et al., 2023] using the MIMIC II annotated arrhythmia dataset [Saeed et al., 2011, Aboukhalil et al., 2008] faced challenges when applied to the VTaC dataset. In particular, the diffusion model with contrastive learning [Wu et al., 2023] outperformed other baselines when applied to the MIMIC dataset but under-performed with the current VTaC dataset. The performance gap can be attributed to several potential factors. Firstly, the imbalanced label distribution in VTaC, with a relatively lower rate of true alarms at 28.6%, in comparison to over 50% true alarms in the MIMIC II dataset, presents a potential limitation, particularly for generative approaches proposed in [Wu et al., 2023], which depended on training

with abundant samples of true positives. Secondly, less than 40% of alarm records in VTaC included arterial blood pressure (ABP) waveforms, in contrast to the MIMIC annotated arrhythmia dataset, which had a higher percentage of ABP records due to its specific sampling criteria. Finally, the MIMIC II arrhythmia dataset is a single-center, relatively homogeneous dataset collected from patient monitors of a single monitoring system, whereas VTaC is a multi-center dataset.

## 7 Discussion

Previous algorithm development in false arrhythmia reduction has been hampered by the use of small, single-center, or relatively homogeneous datasets. This limitation hinders their generalizability and real-world applicability. The introduction of this large-scale annotated VT alarm dataset provides a valuable opportunity to address this challenge. The VTaC dataset presented in this study, encompasses data collected from ICUs in three major US hospitals and incorporates data from three major bedside monitor manufacturers. By encompassing data from diverse sources, this dataset enables the evaluation and refinement of algorithms in a broader context, across a more diverse monitoring devices and clinical settings. Furthermore, the random selection of VT alarm events from the available data for annotation ensures that the dataset's distribution and the proportion of true alarms closely resemble those observed in real ICU settings. This approach enhances the dataset's representativeness and allows for a more accurate evaluation of the models' performance in practical clinical scenarios.

**Safety & Ethical Discussion** We raise several safety and ethical considerations. Firstly, as the dataset contains waveform recordings from real patients, we acknowledge the importance of patient privacy and data protection. To ensure privacy, we have de-identified the data by removing any personally identifiable information. Secondly, the process of labeling the dataset involved clinical experts reviewing waveform recordings and making labeling decisions. We recognize the potential influence of individual subjectivity and variability among annotators. To mitigate this, we have sought to ensure robustness by obtaining labeling decisions from multiple experts for each alarm event. This approach helps to capture diverse perspectives and minimize potential bias in the annotations. We have made efforts to curate a diverse and representative dataset, but there might still be underlying biases that could impact the performance of the machine learning models.

**Limitations & Future Work** The newly introduced annotated VT alarm database is specifically designed to tackle the challenges related to reducing false VT alarms. The VT alarm events are randomly sampled, and thus the selected VT events cannot be viewed as a comprehensive collection of all VT alarm events for an individual patient. As such, the database is not designed for long-term forecasting of arrhythmia episode onset. Another limitation of our study is the absence of detailed clinical information accompanying the waveform recordings. We leave the collection of matched clinical data for future research endeavors. Finally, while the biases from expert annotators in this dataset is greatly reduced from having multiple annotators, we acknowledge that biases can still be present if two or more annotators have the same bias.

In our analyses, we have opted to construct our machine learning models by utilizing data from up to 10 seconds before the alarm onset (except the contrastive learning based approaches, which sample data from prior to 10 seconds before the alarm onset). Additionally, this study involved the inclusion of a maximum of four channels of available waveforms per event to formulate our models. For future investigations, we aim to explore models capable of more effectively handling higher-dimensional multi-channel waveform data encompassing longer sequences. This will enable us to fully leverage data from earlier time points and identify patterns across multiple channels of physiological waveforms.

## 8 Conclusion

Ventricular tachycardia is the most challenging arrhythmia to detect reliably, and remains a continuing problem in ICUs despite decades of effort from industrial and academic algorithm developers. We present a new annotated VT database to address the challenges associated with reducing false VT alarms. We conducted a comprehensive benchmarking of various machine learning models utilizing this annotated VT dataset. By providing this resource as an open-access database, freely available to the research community, we aim to foster collaboration, facilitate benchmarking, and encourages the advancement of research efforts in the field of arrhythmia alarm analysis.

## ACKNOWLEDGMENTS

This study was funded by NIH grant R01EB030362. L Lehman was in part funded by MIT-IBM Watson AI Lab. The authors would like to express their gratitude to Dr. Eric Gottlieb and Dr. Xavier Borrat Frigola for their contributions to the annotations. Additionally, we extend our appreciation to Shaherul Haque and Anthony Baez at MIT for their technical assistance.

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

# Appendix A  Additional VT Alarm Dataset Description

This dataset is a compilation of patient waveforms sourced from multiple institutions, and the process of collecting this data has received approval from the respective Institutional Review Boards (IRBs) of each participating institution. Requirement for individual patient consent was waived because the project did not impact clinical care and all protected health information was de-identified. It is important to emphasize that the alarms and waveform records included in this dataset encompass data collected up until 2022 and do not overlap with those of the PhysioNet Challenge 2015.

Continuous multi-channel waveform recordings were collected from ICUs of three major US hospitals. The dataset consists of a total of 20,158 continuous waveform recordings collected between 2013 and 2022, and represents data collected from three major bedside monitor manufacturers. The average duration of the patient waveform recording is 4.5 days. We randomly extracted recordings with at least one VT alarm. The resulting dataset consists of a total of 18,465 waveform alarm events, sourced from a randomly selected 2,376 unique patient waveform recordings (with at least one VT alarm). The annotated VT alarm events (5,742 alarms from 2,376 patient records) were randomly drawn from this collection, with the constraint that no more than a maximum of five VT alarms can be sampled from the same patient waveform records to ensure diversity. After removing confirmed Reject/Uncertain events (by at least two independent annotators), 5,037 alarm events (from 2,260 unique patient records) remain for algorithm training and evaluation. Table 5 displays the number of events receiving unanimous vs conflicting decisions respectively. Unanimous category represents events with decisions from two independent annotators in agreement. Figure 2 illustrates the VTaC data collection and annotation process.

Table 5: Annotated VT alarm events categorized by annotation agreement.

|  | Unanimous (79%) | Conflicts (21%) | | Total |
|---|---|---|---|---|
|  |  | Adjudicated | Un-Adjudicated |  |
| Alarm Events | 4,534 | 816 | 392 | 5,742 |
| Labeling Decisions | 4,534x2 | 816x3 | 392x2 | 12,300 |
| Events with Final Decisions | 4,534 | 816 | 0 | 5,350 |

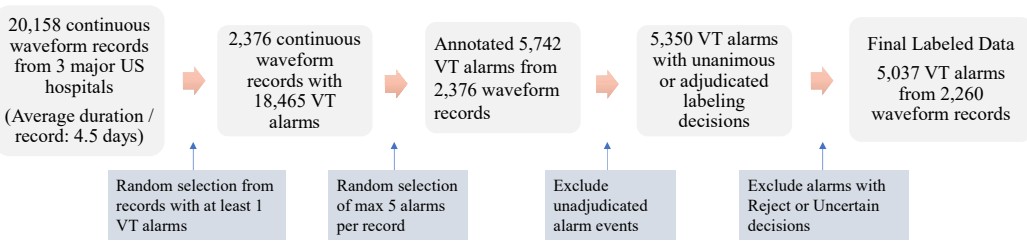

Figure 2: VTaC data collection, event sampling and annotation pipeline.

ECG and pulsatile signals were recorded by the bedside monitor at various sampling frequencies ranging from 100 Hz to 250 Hz. ECG signals were recorded at a minimum of 240 Hz. ABP and PPG were all sampled at the same rate which would be either 100 / 125 / 240 Hz. All of the signals were resampled to 250 Hz for consistency.

Among the annotated 5,037 VT events, all events have at least two channels of ECG, and 36% and 91% of those events also had ABP and PPG respectively. The median number of ECG leads per waveform recording is 3. In Table 6, we show the summary statistics of the number of ECG leads per waveform recording in the dataset.

# Appendix B  Annotation Tool and Team

Annotation was performed using an open-source annotation platform, PhysioTag, which enables experts to collaboratively annotate physiological waveform records using a standard web browser.

Table 6: Summary statistics for the number of ECG leads per waveform recording in the VTaC dataset.

|  | # of ECG Leads |
| --- | --- |
| Average | 3.4 |
| Median | 3 |
| 25% percentile | 3 |
| 75% percentile | 4 |

Table 7: Real-Time event: Test set performance of the model that exhibited the best validation performance among the 10 random seeds. Top-performing models for each specific metric highlighted in bold. CL=Contrastive Learning.

| Methods | | TPR | TNR | PPV | F1 | Score | AUC |
| --- | --- | --- | --- | --- | --- | --- | --- |
| Rule-based | | 0.942 | 0.629 | 0.502 | 0.655 | 67.32 | – |
| Supervised | MLP | 0.708 | 0.562 | 0.391 | 0.503 | 45.33 | 0.703 |
| | SAE | 0.847 | 0.803 | 0.630 | 0.723 | 69.43 | 0.905 |
| | Transformer | 0.796 | 0.719 | 0.529 | 0.636 | 60.10 | 0.844 |
| | CNN | 0.927 | 0.788 | 0.635 | 0.754 | 76.43 | 0.936 |
| | FCN | **0.949** | **0.835** | **0.695** | **0.802** | **81.96** | **0.948** |
| Supervised + CL | CNN+CL | 0.912 | 0.835 | 0.687 | 0.784 | 77.92 | 0.946 |
| | FCN+CL | 0.912 | 0.829 | 0.680 | 0.779 | 77.55 | 0.922 |
| Generative | BeatGAN | 0.803 | 0.473 | 0.377 | 0.513 | 46.27 | 0.638 |
| | TAnoGAN | 0.723 | 0.580 | 0.406 | 0.520 | 47.16 | 0.651 |
| | Diffusion+CL | 0.883 | 0.499 | 0.412 | 0.561 | 53.66 | 0.691 |

The software is freely available on PhysioNet (`https://physionet.org/content/physiotag/`). Events are assigned to annotators dynamically. When a new annotator joins the project, they are assigned 100 randomly-selected events that have not yet been annotated. After they have finished reviewing the assigned events, they can assign themselves a new batch of events. For our VT annotation task, users were given the options of "True" for when they believe the alarm was correct, "False" for when they believe the alarm was incorrect, "Uncertain" for when they are unsure which annotation to assign, "Reject" for when the alarm is unreadable due to noise, artifacts, or other hindrance, and "Save for Later" for when the user would like to return to annotate this event at another time. For more detailed description of the annotation platform for this project, please see [McCullum et al., 2023].

The annotation team consists of six annotators, including a leading arrhythmia analysis expert physician who built the MIT-BIH Arrhythmia database, and a highly-experienced board certified cardiac arrhythmia technician. The remaining three clinical annotators had the following expertise: an endocrionologist and internist with extensive experience in Emergency Medicine and cardiovascular intensive care; an inpatient physician, board certified in nephrology and internal medicine, with substantial experience managing patients in critical care and telemetry settings; and an anesthesiologist working in an ICU setting. The annotation team also included a senior biomedical research engineer with decades of experience in arrhythmia analyses.

## Appendix C   Additional Results on Model Performance

### C.1   Real-time event

In this section, we present additional results for the real-time VT alarm classification task. Table 7 displays the test set performance of the model that exhibited the best validation performance among the 10 random seeds. Model selection was based on the best Challenge Score on the validation set. In Figure 3 we plot the ROC curves and their operating points for a subset of the models in the real-time event.

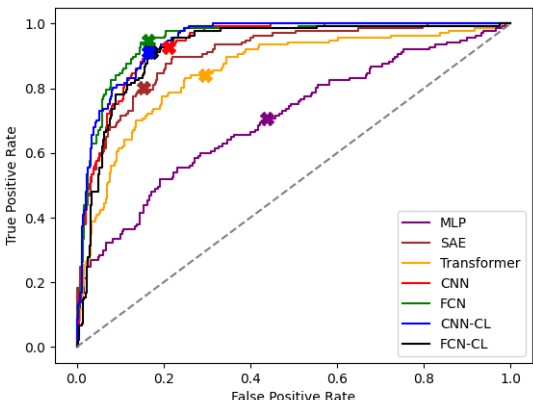

Figure 3: ROC curves and their operating points for a subset of models in the real-time event. Operating point (marked as dots) for each model on the ROC curve was selected using the threshold that resulted in the best validation Challenge Score.

Table 8: Retrospective event performance from the VTaC database. Mean and standard deviation from 5 seeds shown. Top-performing models for each specific metric highlighted in bold. CL=Contrastive Learning.

| Methods | | TPR | TNR | PPV | F1 | Score | AUC |
|---|---|---|---|---|---|---|---|
| Rule-based | | **0.964** | 0.571 | 0.471 | 0.633 | 65.54 | – |
| Supervised | MLP | $0.585_{\pm 0.064}$ | $0.711_{\pm 0.162}$ | $0.445_{\pm 0.168}$ | $0.505_{\pm 0.034}$ | $46.02_{\pm 2.85}$ | $0.695_{\pm 0.016}$ |
| | SAE | $0.836_{\pm 0.046}$ | $0.761_{\pm 0.053}$ | $0.585_{\pm 0.043}$ | $0.687_{\pm 0.022}$ | $66.01_{\pm 2.37}$ | $0.878_{\pm 0.0159}$ |
| | Transformer | $0.802_{\pm 0.075}$ | $0.677_{\pm 0.066}$ | $0.500_{\pm 0.031}$ | $0.613_{\pm 0.007}$ | $58.20_{\pm 2.07}$ | $0.811_{\pm 0.029}$ |
| | CNN | $0.923_{\pm 0.020}$ | $0.803_{\pm 0.025}$ | $0.651_{\pm 0.026}$ | $0.763_{\pm 0.015}$ | $76.94_{\pm 1.61}$ | $0.942_{\pm 0.003}$ |
| | FCN | $0.923_{\pm 0.012}$ | $0.786_{\pm 0.025}$ | $0.632_{\pm 0.025}$ | $0.750_{\pm 0.016}$ | $75.84_{\pm 1.86}$ | $0.925_{\pm 0.014}$ |
| Supervised+CL | CNN+CL | $0.920_{\pm 0.023}$ | **$0.837_{\pm 0.021}$** | **$0.692_{\pm 0.025}$** | **$0.789_{\pm 0.014}$** | $78.85_{\pm 1.95}$ | **$0.946_{\pm 0.007}$** |
| | FCN+CL | $0.921_{\pm 0.013}$ | $0.836_{\pm 0.011}$ | $0.691_{\pm 0.013}$ | $0.789_{\pm 0.001}$ | **$78.95_{\pm 1.41}$** | $0.933_{\pm 0.007}$ |
| Generative | BeatGan | $0.781_{\pm 0.093}$ | $0.497_{\pm 0.114}$ | $0.385_{\pm 0.023}$ | $0.513_{\pm 0.013}$ | $46.22_{\pm 1.77}$ | $0.639_{\pm 0.017}$ |
| | TAnoGAN | $0.749_{\pm 0.051}$ | $0.546_{\pm 0.056}$ | $0.397_{\pm 0.014}$ | $0.518_{\pm 0.004}$ | $46.97_{\pm 0.49}$ | $0.648_{\pm 0.005}$ |
| | Diffusion+CL | $0.630_{\pm 0.183}$ | $0.503_{\pm 0.046}$ | $0.462_{\pm 0.069}$ | $0.518_{\pm 0.038}$ | $49.70_{\pm 2.42}$ | $0.569_{\pm 0.093}$ |

## C.2 Performance in Retrospective Event

In the context of the retrospective event, machine learning models were assessed for their ability to reduce false arrhythmia alarms using not only the data preceding the alarm but also five seconds of data after the alarm. For retrospective events, we used the optimal hyper-parameter setting from the grid search of the real-time event, and trained the model with that setting using 10 different random seeds. Table 8 displays the performance averaged over the models from the 5 random seeds with the best validation performance.

The winning entry of the 2015 PhysioNet challenge using a Rule-based approach achieved a Challenge Score of 65.54 in the retrospective event. The CNN and FCN models and their contrastive learning variants remained consistently strong performers in the retrospective event. In comparing the performance of the real-time and retrospective events, FCN with contrastive learning had noticeable improvement in the challenge scores. Other models did not benefit significantly from the additional five seconds of post-alarm data as input to the model.

# Appendix D Machine Learning Model Setting

## D.1 Supervised Models

Table 9 displays the range of hyperparameter settings employed during the grid search as part of the hyperparameter tuning process. To overcome the problem of imbalanced classes, a positive class weight was added to the positive samples in the BCE loss function.

The Multilayer Perceptron (MLP) consists of three hidden layers with node sizes of 1024, 256, and 64, respectively. The Supervised Autoencoder (SAE) architecture utilizes an encoder and decoder, both composed of four convolutional layers. A four layer MLP operates on the 32 dim vector to produce model predictions. Learning rate, dropout probability and positive class weight were chosen from hyperparam tuning via grid search based on best validation set performance.

The Convolutional Neural Network (CNN) architecture employed in this study consists of four 1-dimensional convolutional feature extractors. Each extractor is composed of two convolutional layers with window sizes of 50, 100, 200, and 400, respectively. Dropout regularization is applied to mitigate overfitting. The resulting feature maps are then flattened and passed through a hidden layer with 128 nodes before being used to make the model's final prediction. The FCN used 3 convolution layers followed by an adaptive pooling layer. Between each convolution, batch normalization was used. Lastly, we used adaptive max pooling followed by a dense layer which is used to make the model's final prediction.

The Transformer model consists of 1 single-layer Transformer encoders. The dimensionality of the representations is set to 128, and number of attention head is set to 8. The input time steps are first mapped to a 128-dimensional feature space using a linear layer. The processed features are then passed through the Transformer layers and activated using the GELU activation function. The Transformer model use dropout with 0.1. The models are trained using BCE loss as the loss function and please refer to Table.9 for specific hyperparameter settings.

Code is available at `https://github.com/ML-Health/VTaC/`. Code for the rule-based approach [Plesinger et al., 2015] is from the Challenge 2015 PhysioNet website [2].

## D.2 Conditional Generative Models

In addition to conventional classification models, we also employed conditional generative models [Wu et al., 2023, Zhou et al., 2019, Bashar and Nayak, 2020] to classify the VT alarms as true or false. The generative models generate waveform trajectories conditioned on a patient's observed waveform data. Specifically, conditioned on an observed waveform segment from a patient, our generative approach predicts what the subsequent trajectory of the patient's waveform would look like if it were a true alarm. Thus, in the case of a false alarm, the predicted waveform (simulating a true alarm) will differ significantly from the observed ground truth. The distance between the generated trajectory and the original waveform is thus calculated to determine the type of alarm. Since false alarms can be caused by various factors, while true alarms have specific pathological features, we choose to generate trajectories for true alarms to calculate the anomaly scores. If the candidate observed sample is from a genuine arrhythmia alarm, the distance between the generated waveforms and the observed samples will be small; on the other hand, if the alarm is false, the discrepancy between the two will be large. Unlike the classification models, we need to generate physiological waveforms using time intervals that correspond to the true event as much as possible. For BeatGAN [Zhou et al., 2019] and TanoGAN [Bashar and Nayak, 2020], We used their Github repository's settings[3] [4] and modified the input channels to fit the number of channels in our dataset, and we changed the threshold setting from a fixed threshold to a threshold search approach using validation data. In the BeatGAN and TanoGAN models, the learning rate is set to 0.0001, the weight decay is set to 0.0005, and the batch size is 128.

The diffusion-based generative model consists of a network of 36 residual layers with 256 residual and skip channels. The diffusion embedding layer have three level of diffusion embedding of 128, 256, and 256 dimensions. Each layer are connected by a swish activation function. Then, we leverage

---

[2]https://www.physionet.org/static/published-projects/challenge-2015/1.0.0/sources/

[3]https://github.com/hi-bingo/BeatGAN

[4]https://github.com/mdabashar/TAnoGAN

two Transformer encoders for extracting the noise input and conditional input. Each encoder contains one encoder layer and the "dmodel" of each encoder layer is 512 and "nhead" is 4, and feed forward layer between each encoder layer have 512 dimensions. The number of negative samples is 32 and the temperature parameter of total loss $lamda$ is 0.5. In the inference stage, we used 200-time steps on a linear schedule for diffusion configuration from a $beta$ of 0.0001 to 0.02. We utilize an Adam as the optimizer with the learning rate of 1e-4. We randomly mask the half part of data for train and mask the first half of data for test. Because of the $x_{noise} \sim \mathcal{N}(0, I)$ where $I = 1$, we scaled down the ABP channel by a factor of 10 to solve the problem of sample value out of range. We use the MSE Loss to calculate the distance between the generated results and the original waveforms. The threshold used for the test set is the one corresponding to the optimal score obtained on the validation.

Table 9: Model Hyperparameters and their Search Range. Parameter settings with the best validation Score underlined.

| Hyperparameters | Search Range |
|---|---|
| **MLP** | |
| Batch Size | 32, 64, 128 |
| Learning Rate | 0.0001, 0.001 |
| Dropout Probability | 0.0, 0.1, 0.3 |
| Positive Class Weight | 3.54, 4 |
| **SAE** | |
| Batch Size | 32, 64, 128 |
| Learning Rate | 0.0001, 0.001 |
| Reconstruction Loss Weight | 0.5, 1.0, 1.5 |
| Dropout Probability | 0.0, 0.1, 0.3 |
| Positive Class Weight | 3.54, 4 |
| **CNN** | |
| Batch Size | 32, 64, 128 |
| Learning Rate | 0.0001, 0.001 |
| Dropout Probability | 0.0, 0.1, 0.3 |
| Positive Class Weight | 3.54, 4 |
| **CNN + CL** | |
| Batch Size | 32, 64, 128 |
| Learning Rate | 0.0001, 0.001 |
| Dropout Probability | 0.1, 0.3, 0.5 |
| Contrastive Loss Weight | 0.5, 1.0, 1.5 |
| Positive Class Weight | 3.54, 4 |
| **FCN** | |
| Batch Size | 32, 64, 128 |
| Learning Rate | 0.0001, 0.001 |
| Dropout Probability | 0.0, 0.1, 0.3 |
| Positive Class Weight | 3.54, 4 |
| **FCN + CL** | |
| Batch Size | 32, 64, 128 |
| Learning Rate | 0.0001, 0.001 |
| Dropout Probability | 0.1, 0.3, 0.5 |
| Contrastive Loss Weight | 0.5, 1.0, 1.5 |
| Positive Class Weight | 3.54, 4 |
| **Transformer** | |
| Batch Size | 32, 64, 128 |
| Learning Rate | 0.0001, 0.001 |
| Dropout Probability | 0.0, 0.1, 0.3 |
| Positive Class Weight | 3.54, 4 |

### D.3 Hardware and Software

In our experiments, we used NVIDIA Tesla V100 GPUs with 32 GB of VRAM and IBM Power9 CPUs, running on Red Hat Enterprise Linux 8.3.1. We used Python version 3.8.18, Pytorch version 1.9.0 using a CUDA version 12.2 backend and WFDB version 4.1.1.

