# OpenReview forum: "VTaC: A  Benchmark Dataset of Ventricular Tachycardia Alarms from ICU Monitors"
_NeurIPS.cc/2023/Track/Datasets_and_Benchmarks — NeurIPS 2023 Datasets and Benchmarks Poster_

### Official Review · Reviewer_sRHE · 2023-07-21
**A Large Scale Annotated Dataset of Ventricular Tachycardia Alarms from ICU Monitors**

**Rating:** 7
**Confidence:** 5
**Clarity:** well written paper

**Strengths:**

The availability of a large annotated database for Ventricular Tachycardia Alarms highlights the complexity of the task and emphasizes the necessity for ongoing research in this field. As a result, it remains an open and challenging problem from the perspective of machine learning.

**Additional Feedback:**

It is good paper, focusing on very challenging medical problems. Some good figure in paper will enhances the readability more.

**Correctness:**

Claims made in the submission are correct, and have many relative findings to support. Benchmark experiment are designed in good way, and also evaluation process.

**Documentation:**

There are sufficient details available  on data collection and organization, and maintenance, and ethical and responsible use. Data link is working fine, and showing data waveform very smoothly.

Regrading  reproducibility for benchmark, Github not shared. Based on information shared in Supplemental Material, Code for the machine learning models used in this study will be made available upon publication of the paper

**Ethics:**

I found no ethical concerns.

**Limitations:**

In paper, limitations has already been discussed. However, i would like to know more on your saying: absence of detailed clinical information accompanying the waveform recordings, and  following sentence.

**Opportunities For Improvement:**

Apart from detailing of dataset covering various aspects, it would be great if pictorially representation can be added to easily follow the paper. I missed it here.

Good to add more on how to dealing with unbalancing problem; for traditional and CNNs.

**Relation To Prior Work:**

proposed database is compared with various existing database for same problem. Also discussed related work with good understanding.

**Summary And Contributions:**

This paper proposed a largely annotated dataset along with baseline results;  for ventricular tachycardia (VT) (know for most challenging arrhythmia to detect reliably) alarm database. The dataset includes 10,939 labeling decisions made by six experts who reviewed 5,759 waveform recordings from patient bedside monitors in an Intensive Care Unit (ICU) setting. The recordings were obtained from three commercial vendors and originated from multiple ICUs in two major hospitals in the United States, offering a diverse and representative collection of waveform data. Each waveform recording contains electrocardiogram (ECG) leads and one or more pulsatile waveforms, such as photoplethysmogram (PPG or PLETH) and arterial blood pressure (ABP) waveforms.

Contribution:(1) compared with other existing database,  it includes labelling decisions from multiple independent human annotators for a majority of the records, which further helps to ensure the robustness by reducing the potential impact of individual subjectivity and variability among annotators. (2) They have showed very comprehensive benchmark of multiple machine learning models using proposed annotated VT dataset. (3) large number of recordings.

---

> ### Author Response · Authors · 2023-08-27
>
> Thank you for your valuable review feedback and insightful comments. We greatly appreciate your thoughtful suggestions. Please see below for our point-by-point response.
>
> Visual representation: Thank you for your suggestion!  We have added illustrative examples of the annotated waveform alarm data in the Appendix B and will add additional visual representations to illustrate our approach in the final version of our paper.
>
> Dealing with Unbalanced Data: For deep learning approaches, we adjusted the weight for positive examples in the binary cross entropy loss (i.e., pos_weight parameter in the BCEloss) to handle the effects brought about by imbalanced sample labels. pos_weight allows us to assign a weight to the positive class (label 1), helping the model to give more attention to the minority class. For instance, if there are far fewer positive samples than negative ones in the dataset, pos_weight can be set higher than 1 to emphasize the importance of correctly classifying positive samples. Specifically, different deep learning methods have distinct parameter settings, which can be found in the Appendix. In our experiments, we generally set the pos_weight parameter to 4, reflecting the relative proportion of true vs false alarms in the data.
>
> Need for detailed clinical information:  Regarding the need for detailed clinical information, we note that algorithm development in false arrhythmia alarm detection has primarily focused on using multi-channel waveform recordings from real-time patient monitors as input to the models. While we agree that detailed clinical information would provide additional context, we believe that the introduced large-scale annotated VT multi-channel waveform recordings sourced from multiple commercial patient monitors across two major US hospitals provides a valuable, diverse real-world database  that both commercial and academic researchers can use to develop more robust and accurate algorithms for false VT alarm reduction.
>
> Regarding the challenges in obtaining mapped clinical information, we are not able to provide the clinical information for these patients at this point, as we do not have access to the complete linkage between the collected waveforms and the corresponding clinical information.   In order to establish the linkage, the hospitals would need to provide reliable and precise linkages between patient bed numbers (which waveforms and monitoring events are associated with) and the patient identifiers in the clinical database.  Such linkages, in our experience, can be noisy and unreliable, potentially requiring multiple years of effort to establish accurately. We have on-going efforts to match some of these physiological waveform records to clinical data.  We leave the release of such mapped clinical data to waveform recordings to future work.
>
> We are grateful for your valuable review feedback and suggestions, which have helped improve our work.

---

### Official Review · Reviewer_uV7r · 2023-07-21
**A new ventricular tachycardia dataset with extensive tests but is missing cohort demographic information**

**Rating:** 6
**Confidence:** 4
**Clarity:** The paper is very well written.

**Strengths:**

- Dataset will be publicly accessible (in June 2024)
- Quite high agreement on annotations, suggesting the labels are high quality
- Well written
- Tests a range SoTA approaches on the dataset

**Additional Feedback:**

- Table on line 199 extends out of the text margin. Consider wrapping the wide columns.
- Similar issue with table 5, but unclear what the best to solve that issue is.
- It would be interesting to see a model trained on unanimous decision only, as this is hypothetically the data with the clearest signal, and then test is on the same test set as used here.

**Correctness:**

The dataset is missing some key demographic information which will help estimate the bias. Hospital data is almost always skewed, but demographic information helps users of the dataset be aware of the biases, which is critical.

**Documentation:**

This is the biggest drawback of the paper. The lack of documentation surrounding the study, collection and demographics. The most detail I find is that it includes "recent data collected up until 2022". Important information includes, but is not limited to, the study/collection design, population demographics, eligibility criteria, biases, any removal of data and the decision criteria.

I recommend the authors look at the STROBE Checklists, or similar, for inspiration on how to report these meta data.

**Ethics:**

Yes, the authors have discussed some of the ethical factors. Mostly from a privacy perspective.

However, description of the consent mechanism for the study and its limitations (if any) are missing.

I don't think warrants a special ethics review, but a statement on patient consent should be required before publication.

**Limitations:**

The authors do discuss the limitations of their work, with a particular focus on patient privacy. Some more discussion on limitation is always welcomed. One limitation are the labelers (note, this is not a problem with the paper, but might be worth acknowledging), their biases which is greatly reduced by having multiple labelers, can still be present if two or more labelers have the same bias.

**Opportunities For Improvement:**

There are some clear and critical omissions:
- The required checklist is missing
- A description of the population demographics of the participants is missing. This is critical for estimating potential biases in the dataset.

**Relation To Prior Work:**

Yes, the paper discusses prior work and establishes connection with previous datasets (e.g. in physionet) and compares methods that have been developed on related databases.

**Summary And Contributions:**

Here, the authors present an annotated dataset of VT (ventricular tachycardia) alarms collected from hospital bedside monitors. The recordings have been annotated by experts (often with two experts per datapoint). The dataset is similar to the physionet 2015 competition data but includes newer data.

The authors then train and test a number of different models on the new data and test their performance. The results indicate that the SOTA models can perform reasonably well on the detection of VT.

---

> ### Author Response · Authors · 2023-08-27
>
> Thank you for your valuable review feedback. We greatly appreciate your thoughtful suggestions. Please see below for our point-by-point response.
>
> Additional documentation: We have added additional documentation to the Appendix to detail the waveform data collection, preparation, and statistics on signal distributions. We have also added a statement regarding patient consent in the Appendix B dataset description section.  While we are not able to provide demographics information (see below), we note that we provided detailed account on other aspects of the dataset and the study, including the annotation process, and individual-level labeling decisions at a much greater granularity than other existing labeled dataset to our knowledge. We believe that such granular information on labeling decisions introduces opportunities for new research, specifically in understanding model performance as a function of annotator agreement.  These aspects are typically overlooked by other studies and conventional checklists.
>
> We thank you for your suggestion to add statements regarding potential bias from the annotators; we have added these statements in Section 8 (Limitation & Future Work) in the main text.
>
> Additional Experiments: Following your suggestions, we are conducting experiments to test if a model trained on unanimous decision only would provide enhanced performance due to clearer signals in the training data. We have added the experimental results in the Appendix.  Thank you for your suggestion.
>
> Need for Detailed Clinical Information: We agree that detailed clinical information would provide additional context and important information. However, we are not able to provide the clinical information for these patients at this point, as we do not have access to the complete linkage between the collected waveforms and the corresponding clinical information.  Establishing such linkage is a multi-year effort. Our team has been working on matching some of these physiological waveform records to clinical data and will leave the release of such mapped clinical data to waveform recordings to future work. (Please also see our response to reviewer sRHE21)
>
> We note that algorithm development in arrhythmia alarm detection has primarily focused on using multi-channel waveform recordings from real-time patient monitors as input to the models.  Landmark arrhythmia databases like the MIT-BIH Arrhythmia Database have made a profound impact, enabling arrhythmia research that was not possible before. While we acknowledge the significance of investigating algorithm bias with additional clinical information, we want to underscore the invaluable contribution of a meticulously curated, openly accessible, multi-center waveform database carefully annotated by highly skilled clinical experts. Such resources are pivotal in advancing algorithmic solutions to address long-standing, unresolved challenges in healthcare.  We believe that the introduced large-scale annotated VT multi-channel waveform recordings sourced from multiple commercial patient monitors across two major US hospitals provides a valuable, diverse real-world database that can be used to develop more robust and accurate algorithms to address a longstanding and unsolved challenge in ICUs.
>
> Table lengths and formatting: Thank you for raising this point. The table length issue is now resolved.
>
> We are grateful for your valuable review feedback and helpful suggestions.

---

### Official Review · Reviewer_2kri · 2023-07-21
**VT alarm database including labeling from clinical experts**

**Rating:** 6
**Confidence:** 4
**Clarity:** This paper is highly well-written and…

**Strengths:**

In intensive care units, false arrhythmia alarms are problematic as it is hard to reliably detect VT and there are so many alarms. This dataset is the largest open-access, federated VT alarm dataset, annotated by clinical experts. Some annotators are highly skilled clinical experts related to the targeted disease.

While collecting the dataset, the authors appropriately exclude uncertain or conflicted events. Even though not every record is examined by more than two annotators, the authors analyze the annotator's agreement.

The experiments include a reasonable number of baseline models and the authors also report the performance of the recent models.

**Additional Feedback:**

This work was easy to understand and very well written, but I wish it had been more extensive.

**Correctness:**

The claims made in this paper are correct. This dataset is constructed in a sound way.

**Documentation:**

There is sufficient detail on data collection.

**Ethics:**

The authors discuss safety and ethics. It might have been better if there is demographic information of the original data as it is important to have unbiased patient samples.

**Limitations:**

The authors clearly mentioned that the database is not for predicting arrhythmia.

**Opportunities For Improvement:**

As this is the dataset where "whether the alarm is false or not" is labeled, it can serve more as reducing the fatigue of false alarms than improving the performance of the arrhythmia detection itself. Also, the authors only concentrate on the "VT ALARM" case, so it might have been better if other related cases are included in the dataset. This seems to be concentrating on quite a narrow scope.

Even though some annotators are highly skilled clinical experts, it might have been better if there are at least three annotators for each record in order to deeply analyze the agreement among them.

The authors also highlight the huge scale of this dataset (e.g., 10,939 labeling decisions), but the meaningful event is actually 5,296 records. Among them, 998 events are hard to be served as significant results because only one annotator has made the decision.

**Relation To Prior Work:**

The authors clearly discussed how this dataset differs from the previous dataset. It is quite surprising that there has been no related dataset for quite a long time.

**Summary And Contributions:**

Summary
- The paper addresses VT alarm database, sourced from multiple ICUs including labeling from six clinical experts. The label is that whether the VT alarm is false or not. Over 80% of recordings are labeled by at least two annotators. Except reject/uncertain events and conflict events, there are 5,296 final events included.

Contribution
- The dataset is the largest open-access, federated VT alarm dataset, annotated by clinical experts.
- In contrast to the previous dataset, the paper provides fine-grained pre-adjudication labeling decisions from human experts.
- This dataset can work as a good resource to prevent false arrhythmia alarms.

---

> ### Author Response · Authors · 2023-08-27
>
> We greatly appreciate your valuable review feedback.  Please see below for our point-by-point response.
>
> Scope of the Dataset: We focused on the VT alarm case, because it is by far the most challenging life-threatening arrhythmias to detect reliably and presents an unsolved real-world problem in critical care settings. We believe the presented dataset provides a unique and valuable resource that can be used to enable development of high-impact algorithms with significant translational potential, in addressing a critical real-world challenge routinely encountered in intensive care unit settings.  We underscore the value of a meticulously curated, openly accessible, multi-center waveform alarm database carefully annotated by highly skilled clinical experts. To the best of our knowledge, the introduced dataset represents the largest open-access, federated VT alarm dataset.
>
> Expansion of Annotated Data and Proportion of Events with Single-Decision: We have expanded the number of annotations on this dataset.  The single-decision events have now been annotated by an additional independent annotator, so that all events are annotated by two or more independent annotators (with over 12,250 labeling decisions for 5,759 alarm events with waveform recordings).  The public release of this dataset will include the expanded annotations.  Updated statistics and model performance using this expanded dataset will be incorporated in the final camera-ready version of the paper to reflect this new change.
>
> We are grateful for your valuable review feedback.  We have strived to address your comments and remain committed to enhancing the value of the presented data resource to enable the development of high-impact algorithms with significant translational potential.

---

### Official Review · Reviewer_pRjp · 2023-07-21
**A Large Scale Annotated Dataset of Ventricular Tachycardia Alarms from ICU Monitors**

**Rating:** 7
**Confidence:** 4
**Correctness:** Yes the claims made in the manuscript…

**Strengths:**

Data collection methodology is strong, with a larger number of patients and two hospital sites represented in the dataset. The labelling strategy of having multiple annotators for most of the labelled waveforms in the dataset is a strong attribute of this dataset, and will be useful to assess the maximum performance one can aspire to reaching. The total number of labelled waveforms in this dataset is larger as compared to the existing datasets in this area. In all, this dataset will be a valuable resource for researchers developing algorithms to classify true from false VT alarms.

**Additional Feedback:**

NA

**Clarity:**

The paper is well written for the most part, except a few minor typos and errors.

**Documentation:**

It will be beneficial to add some more documentation, see information of anonymized patient, hospital, and bedside monitor ids in opportunities for improvement section.


**Ethics:**

The authors have described that they have obtained approvals from the Institutional Review Board (IRB) in the supplementary material.

**Limitations:**

It will be useful to denote any details of the bedside monitor software that was used to raise the alarms, and discuss any implication of bedside monitor device alarm software version updates on this dataset.


**Opportunities For Improvement:**

It will be useful if the authors make certain information like information on bedside monitors used available such as the number of patients represented after their filtering process.


It is mentioned three popular bedside monitors are used for collecting waveforms in the dataset, it will be useful to provide the labels of these monitors across the datasets. For each of the measurements,  ids  of these monitors (even if anonymized) in the dataset will be useful to assess if the performance of the classifiers are affected by the monitor type. In case if possible, other information like anonymized hospital id would be useful as well. This would allow one to investigate how these variables affect the classifiers.


There seems to some minor discrepancy in the statistics of the dataset. For example in Table-4 1283 samples are annotated for one-decision in Overall and Train columns, however the caption in the table refers to 993 events with one annotation. Further it is different from the what is mentioned in Table 1 for one decision.


In section 6, it will be useful to justify the choice of windows used for real-events and retrospective events. For the real-event 10 sec of the waveform is used prior to the alarm onset, and for retrospective 10 sec of the waveform prior to alarm onset, and 5 sec after the onsets is used. Why so?


It will be useful to provide the software code used to develop the models described in the manuscript with any configurations used in addition to the links to the original models


The rational for including the 285* Reject/Uncertain Events that are unresolved conflicts in the final dataset (in last row of table 1) is not clear. The text below (line 172) mentions such events are excluded. It will be useful to clarify this.


There are a few minor typos, for e.g. line 182 dataset is misspelt as “dataest”.

**Relation To Prior Work:**

Yes the relation to prior work is discussed well.

**Summary And Contributions:**

The authors have curated a dataset of human labelled ventricular tachycardia alarms and labelled a total of 5759 ventricular tachycardia (VT) ECG waveforms. The datasets were sourced from 2383 unique patient waveform recordings with 18,472 alarm events from three bedside monitors.  The dataset was labelled by a team of 6 medical/domain experts annotators where  80% labelled waveforms were annotated by two annotators. The authors further compared the performance of various algorithms and demonstrated the performance of these algorithms are better on this dataset as compared to how those algorithms did on previous datasets. This dataset will be a valuable resource for algorithm researchers to develop classifiers that can separate true alarms from false alarms VT alarms.

---

> ### Author Response · Authors · 2023-08-27
>
> Thank you for your valuable review feedback and insightful comments. We greatly appreciate your thoughtful suggestions. Please see below for our point-by-point response.
>
> 1. Bedside Monitor Information: The specific vendor identity of individual records, unfortunately, cannot be released and must be anonymized at this stage to conform to our data collection and sharing agreement.  The idea of releasing anonymized identifiers for the bedside monitors and hospitals is an interesting one and we greatly appreciate your thoughtful suggestions. For this current release we are unable to include them as a precautionary measure to minimize the risks of re-identification of data sources but may consider this option for future releases.
>
> Both hospitals and device vendors are often, and quite understandably, reluctant to publicize details about alarms. This has directly and indirectly contributed to the lack of a shared public patient monitoring alarm dataset crucial for algorithm development.
>
> Our goal is to provide a diverse real-world dataset that both commercial and academic researchers can use to develop more robust and accurate algorithms.  In openly sharing this federated alarm dataset, we wish to exercise caution and underscore that our goal is not to compare or critique the performance of any particular hospitals, caregivers, devices, or manufacturers, as such analyses could potentially be misleading when taken out of context. We believe that, through careful anonymization of the sources of our data, this federated, anonymized, multi-center, multi-vendor dataset is a major step forward in providing a unique and valuable resource to enhance generalizability in algorithm development by including alarms from diverse commercial monitors and hospital settings.
>
> 2. Clarification on statistics in Table 1 vs 4: Thank you for raising this issue.  The numbers for the one- decision category in Table 1 and Table 4 differ because in Table 4, we reported an aggregate number after augmenting the original one-decision events with the True/False labeling decisions from the “Unresolved Conflicts” category after removing Uncertain decisions. The rationale is that when the data is noisy, while experienced annotators can distinguish between a true vs a false alarm, annotators who are less experienced would tend to have a higher percentage of uncertain or reject labeling decisions due to noise in the data.  (See also comments below in point 5) Additionally, there was a typo in the caption (993 should be 998), which is now fixed. We will make corrections to our paper to clarify this point.  We apologize for the confusion, and we greatly appreciate your comments.  Finally, we have expanded our annotation effort – in the final public release of our dataset, all events would have received labeling decisions from two or more annotators.  We will update the statistics accordingly to reflect this change for the camera-ready version, if accepted.
>
> 3. Justify choice of window size: We use 10-seconds of waveform prior to the alarm onset as input to our real-time model, as the events that triggered the alarms are within 10-seconds from the alarm according to the ANSI/AAMI standards EC13 Cardiac Monitor Standards [1].  We followed the PhysioNet Challenge 2015 in defining the retrospective events, which permitted the participants to use data up to 30 seconds after the alarm onset for the classification tasks.  We used up to 5-seconds after the alarm onset as input to our model for retrospective events to capture most of the important information while maintaining computational efficiency.
> Reference:
> [1] American National Standard (ANSI/AAMI EC13:2002) Cardiac monitors, heart rate meters, and alarms. Arlington, VA: Association for the Advancement of Medical Instrumentation; 2002.
>
> 4. Clarify the 285 events in the Unresolved Conflicts: In the Unresolved Conflicts events where two annotators disagree, we have excluded the events where two annotators chose the opposite alarm labels (true vs false) respectively. We have also excluded events in Unanimous decisions where both annotators labeled the event as Uncertain or Reject. For the events where one of the annotators is uncertain about the labeling decision (typically due to noise in signals), and the other annotator was able to make a labeling decision on the alarm as either True or False, we have opted to include the labeling decision from the latter annotator in the one-decision category in our Training set in our experiments. The rationale is that when the data is noisy, while experienced annotators can distinguish between a true vs a false alarm, annotators who are less experienced would tend to have a higher percentage of uncertain or reject labeling decisions due to noise in the data.   We should emphasize that these events are included only in the Training data, and not in the Validation or Test sets.
>
> We are grateful for your valuable review feedback and helpful suggestions.

---

> > ### Comment · Reviewer_pRjp · 2023-08-31
> >
> > I appreciate and thank the efforts by the authors to address the review comments.

---

### Decision · Program_Chairs · 2023-09-22

**Decision:**

Accept (Poster)

**Comment:**

This work proposes a new waveform dataset (ECG signals with additional pulsatile signals) for detecting ventricular tachycardia (VT), sourced from multiple ICUs of two hospitals in the US. The dataset contains approximately 5,000 samples with automatic alarm records, as well as expert opinions (i.e. whether the alarm was correct or not) annotated by human experts. A set of diverse models were trained on this dataset to demonstrate its utility, compared to a previous waveform dataset.
While the value of the dataset is solid, the narrow usage scope of the dataset remains a concern. The dataset contains only one label, namely the validity of the VT alarm. Therefore the dataset might not be appealing to the wider audience of NeurIPS. But the pratical/clinical utility of the dataset (which is rare in the field of ML for healthcare) is acknowledged.